# Spent Coffee as a Composite Filler for Wastewater Treatment

**DOI:** 10.3390/ma16031181

**Published:** 2023-01-30

**Authors:** Izabela Kruszelnicka, Michał Michałkiewicz, Dobrochna Ginter-Kramarczyk, Przemysław Muszyński, Katarzyna Materna, Marta Wojcieszak, Kamila Mizera, Joanna Ryszkowska

**Affiliations:** 1Department of Water Supply and Bioeconomy, Faculty of Environmental Engineering and Energy, Poznan University of Technology, ul. Berdychowo 4, 60-965 Poznan, Poland; 2Institute of Chemical Technology and Engineering, Poznan University of Technology, ul. Berdychowo 4, 60-965 Poznan, Poland; 3Faculty of Materials Science and Engineering, Warsaw University of Technology, Wołoska 141, 02-507 Warsaw, Poland

**Keywords:** composite with a natural filler, moving bed technology, MBBR, spent coffee grounds (SCG), wastewater treatment technologies, wettability

## Abstract

Currently composites play an important role in all aspects of engineering and technology, with constantly growing applications. Recently, more attention was focused on natural fillers due to their suitability as reinforcement materials in thermo-plastic matrices which improve the mechanical properties of these polymers. Biofillers are used due to their low cost, high strength rigidity, non-toxicity, biodegradability, and availability. Currently, spent coffee grounds (SCG) are attracting more attention as a natural filler since high amounts of SCG are generated every day (food waste of coffee processing). This study allowed us to determine the long-term effect of activated sludge microorganisms with known technical and technological parameters on the mechanical properties of composites with spent coffee grounds filler. The fittings consisted of high-density poly-ethylene (PE-HD), which was used as the matrix, and a filler based on spent coffee grounds (SCG), which was used as a modifier. It was established that the composition of the composite and its residence time in the bioreactor directly influenced the contact angle value. The shift of the contact angle value is associated with the formation of the biofilm on the tested materials. An increase in the contact angle was observed in the case of all samples tested in the bioreactor, with the lowest values equal to approx. 76.4° for sample A (PE-HD) and higher values of approx. 90° for the remaining composite samples with a coffee grounds filler. The research confirmed that the increased ratio of coffee grounds in the composite results in the increased diversity and abundance of microorganisms. The highest number and the greatest diversity of microorganisms were observed in the case of the composite with 40% coffee grounds after more than a year of exposure in the bioreactor, while the composite with 30% SCG was second. *Ciliates (Ciliata*), especially the sessile forms belonging to the *Epistylis genus*, were the most common and the most numerous group of microorganisms in the activated sludge and in the biofilm observed on the samples after immersion in the bioreactor. The conducted research confirms that the use of polymer composite mouldings with a filler in the form of spent coffee grounds as a carrier allows the efficient increase in the population of microorganisms in the bioreactor.

## 1. Introduction

Tightening legislation regarding the quality of treated wastewater, high investment rates, and operating costs drive the search for new solutions in the wastewater treatment process. Different physical and chemical methods are used for the treatment of wastewater such as biological degradation, ion exchange, chemical precipitation, adsorption, reverse osmosis, coagulation, flocculation as well as photocatalysis [1,2,3]. All these treatment methods are characterized by different performance characteristics as well as different direct impacts on the environment. Among the available solutions, the moving bed technology is an interesting method, which has gained increasing popularity around the world. This biological method of wastewater treatment is based on the formation of a biofilm on free-floating carriers. The fittings are kept in motion by mechanical mixing or aeration [4]. This method possesses numerous advantages, including increased efficiency of the treatment plant, stabilization of its operation during load changes, and the fact that it can operate well in limited areas and does not require sludge recirculation [5]. In order to achieve sustainability and low production costs, the fittings should consist of ecological, easily available, and cheap materials that would facilitate the development of microorganisms. Currently, there is a trend to search for alternatives to commonly used carriers, both in terms of shape and material composition. Therefore, the research area associated with polymer composites with natural fillers is constantly developing. Recently, much attention has been focused on wood-polymer composites (WPC) [6]. The selection of a filler depends on the requirements associated with the practical use of the composite. Guidelines associated with the implementation of the circular economy stimulate the research associated with the management of post-production waste and used products, taking into account the ecological and economic factors. Recently, conventional polymer composites are replaced with biocomposites, which consist of at least one bio-derived or biodegradable component [7,8,9].

Increased application of composites can be observed at a global scale. They find applications in the construction of machines, devices, engineering solutions, in the automotive industry, during the production of sport equipment as well as in biological wastewater treatment technologies, e.g., in MBBRs (Moving Bed Biofilm Reactors). Such systems are based on the moving bed technology in which the development of biomass occurs on elements that move freely in the biological reactor. As a result, the proliferation of microorganisms that purify the wastewater is possible in the entire volume of the biological reactor [10,11].

The moving bed technology is based on the process of immobilization of biomass on elements that are immersed and move freely in the treated wastewater. These elements are characterized by a high specific surface area and their density is similar to that of water, which results in their suspension in sewage. This enables appropriate contact of the biofilm with contaminants present in sewage, which microorganisms use as a substrate and source of energy. The fittings can vary in terms of composition, shape, and size, therefore, they are characterized by a different specific surface area. Most often, biomass carriers are formed using polymers [10,12,13,14]. The thickness of the biofilm and the activity of the formed biofilm depends on the intensity of mixing. Appropriate selection of mixing intensity enables biofilm formation on the carriers subject to shearing forces and allows it to maintain its proper thickness (50 ÷ 300 mm) and high activity [15,16].

The moving bed technology is not only limited to urban wastewater treatment. It can be successfully used to treat hospital wastewater, which is a major source of pharmaceuticals, in the industry to remove benzotriazoles and benzothiazoles, to purify industrial laundry wastewater, which mainly contains surfactants, to treat wastewater from the pulp and paper industry, which is characterized by high COD, BOD, and the presence of toxic chemicals, as well as in household treatment plants [13,17,18,19,20].

In classical approaches to wastewater treatment, the presence of a biofilm is often considered disadvantageous, however, in the MBBR technology, the biofilm plays the most important role. It is a cluster of various microorganisms that settle on the solid surface of the fittings inside the bioreactor. Autotrophic and heterotrophic microorganisms exhibit the ability to form biofilms. The resulting biofilm structure is stabilized by EPS (extracellular polymeric substances), which also protect microorganisms against toxic substances and facilitate biofilm adhesion to the surface. The formation of a biofilm is a multi-stage process, which depends on the properties of living microorganisms and the structure of colonized materials [4,21,22]. Therefore, in order to create appropriate conditions for the development of a biofilm, attention should be paid to the selected carrier (its shape, material, active surface, density, and elasticity), the conditions in the reactor such as mixing speed (low mixing speed causes the biofilm to thicken, while excessive mixing prevents the formation of a biofilm on carriers due to high shear forces) as well as the concentration of oxygen supplied and the sewage load [22]. The application of composites of polymers with natural fillers (NFC-natural fibre composites) is an innovative solution in the MBBR technology, and their popularity is growing at a global scale.

Wastewater treatment based on polymer composites is applied, e.g., for removal of heavy metals from wastewater [9], degradation of petrochemical pollutants, dyes as well as complex organic substances using polymer photocatalysts [23,24], removal of oily substances using a ceramic-polymer composite membrane [25], or for treatment of coloured textile wastewater [26]. These methods have been tested and are most often used in classical wastewater treatment processes.

Currently, in accordance with the guidelines of the circular economy, various post-consumer and post-production wastes should be used to prepare polymer composites. Among several possibilities, spent coffee grounds (SCG) are a promising waste material, the amount of which has been systematically increasing. SCG is a product obtained from the production of instant coffee or the waste left over from the preparation of a ground coffee drink. Coffee grounds are a valuable organic raw material and they contain numerous nutrients, primarily proteins, polyphenols, carotenoids, and organic acids. According to global trends, coffee grounds can be used in agriculture as an addition to fertilizers, for biofuel production, in the cosmetics industry, for the production of biodegradable biocomposites used to prepare biodegradable disposable tableware, reusable tableware, kitchen utensils, food containers, packaging, building materials and accessories used in the automotive industry [27,28,29,30]. Other reports also indicate the use of other coffee wastes, e.g., lignocellulosic coffee silverskin, as a filler for high-density polyethylene (HDPE) composites. However, it should be highlighted that fittings used in the MBBR technology were not previously produced using this material [31,32,33]. The results of SCG studies to date indicate that they adsorb various compounds which may also be present in wastewater. SCGs have been shown to adsorb heavy metals (Cd) [34], antibiotics [35] or dissolved metals (Fe, Al, Ca, Co, Mn, Ni, and Zn) [36]. Therefore, it was proposed to prepare composites using SCG as a filler material. The aim of this study was to examine the functional properties of polymer composites with the addition of a natural filler in the form of spent coffee grounds as fittings in the MBBR technology.

## 2. Materials and Methods

The experiments were carried out using composites which primarily consisted of high-density polyethylene (PE-HD, used as the matrix) Hostalen GD 7255 from Basell Orlen Polyolefins, Płock, Poland; with a melt flow rate MFR = 12 g/10 min. Spent coffee grounds (SCG) were used as modifiers—post-consumer coffee grounds were obtained from Costa Cafe in Warsaw. The fillers were dried in an oven at 105 ± 5 °C for 4 h. The dried waste was milled using a coffee grinder. The obtained SCG particles were characterized by size in the range of 13–890 µm. The median particle size was equal to 221 µm (std dev. 165 µm). The particle size of SCG was determined using a Horiba LA-950 laser particle size analyser based on the LALLS (Low Angle Laser Light Scattering) technique to achieve full geometric and morphological characteristics of the powders. The measurements were carried out with the refractive index specific to the cellulose (1.47) laser setting, which allows for the analysis of the full geometrical and morphological characteristics of powders. Detailed results of SCG research were presented in a different study [37].

The tests were conducted using samples with different SCG content (A, B, C, D), the composition of which is described in Table 1.

Paddle-shaped test specimens were obtained using a two-stage process. In the first step, granulates were produced using the extrusion process, while in the second step, samples were formed by the injection process. The extrusion process was carried out using a Metalchem T-45 single-screw extruder with a classic three-zone screw and a granulating head. The temperature values in the individual zones of the screw extruder were, respectively equal to 140 °C, 160 °C, 150 °C, and 165 °C and kept at 165 °C in case of the granulating head. The extrusion was carried out at the rotational speed of the screws equal to 40 rpm. The granulates with the composition described in Table 1 were produced using composites obtained during the extrusion process. Samples for strength tests were formed based on the granulates from the injection process using the Boy 22 injection molding machine (Figure 1). Prior to the tests, each paddle was marked with a nameplate with an appropriate symbol (A, B, C, D), and each symbol was additionally paired with a number (e.g., A1; A2; B4; B7; C2; C8; D1; D12), depending on the composition of the composite. This allowed us to accurately identify individual samples.

The obtained materials were characterized in terms of their physico-mechanical and microbiological properties. The tests were conducted for several sets of samples which included PE-HD (A) and composites (B, C, D). A single set was not placed in the bioreactor (control) while the remaining sets were introduced into the bioreactor and collected after various periods of immersion in the activated sludge chamber.

### 2.1. Density

The density of the samples was measured using an analytical balance manufactured by RADWAG, type WPA 180/C/1 (Radom, Poland), using the hydrostatic weighing method. The samples were weighed in 96% ethyl alcohol. Five samples for each type of composite were used to determine the density of the samples, and the results were used to calculate the arithmetic mean of these measurements.

### 2.2. Water Content

Prior to placing the samples in the bioreactor, the water content of all materials was checked. Five samples of each type of material were randomly selected for testing. The samples were dried at 60 °C for 2 h in a Memmert dryer, type SF 75, Memmert GmbH + Co.KG, Schwabach, Germany, until a constant weight was achieved. Before the next weighing, the samples were cooled to room temperature (23 °C) in a desiccator with molecular sieves. The tests were carried out for five samples of each type of composite, and the results were used to calculate the arithmetic mean of these measurements.

### 2.3. Water Absorption

Determination of water absorption was carried out based on the ISO 62:2008 [38] standard. Two paddles of each type of material were placed in an oven for 24 h at 50 °C. After this time, the samples were cooled to room temperature in a desiccator. Then, the mass of the paddles was measured and they were placed in a screw cap bottle filled with distilled water in a manner that ensured that each sample was completely immersed in water. The bottles and samples were kept at 23 °C in an incubator from Binder GmbH, Tuttlingen, Germany, for 24 h. After 24 h, the mass of the samples was measured again, and then the percentage change in mass of the sample in relation to the initial mass was calculated on the basis of formula (1) included in the standard:(1)$$c=\frac{m_{2\;}-m_{1\;}}{m_{1\;}}\cdot 100\;\;\left[\%\right]\;$$
where:

*m*_1_—mass of the sample after drying, before immersion in water [mg]

*m*_2_—mass of the sample after immersion in water for 24 h [mg]

### 2.4. Water Contact Angle

Contact angle measurements were carried out using a Drop Shape Analysis System DSA100E (KRÜSS GmbH, Germany, accuracy ±0.01 mN/m). The calculation method (Young-Laplace) is based on the sessile drop method, i.e., drops of liquid are deposited on a solid surface. The drop is prepared prior to the measurement and has a constant volume during the measurement. The Young-Laplace fitting is the most complicated, but theoretically also the most exactly accurate for calculating the contact angle. In this method, the complete drop contour is evaluated. The contoured fitting includes a correction that considers that the shape of the drop does not result solely from interfacial effects, but that it is also distorted by the weight of the liquid it contains. After the successful fitting of the Young-Laplace equation, the contact angle was determined as the slope of the contour line at the 3-phase contact point.

### 2.5. Tensile Strength

Determination of the strength properties of the paddles at static stretching was performed on the basis of the ISO 527-2:2012 [39] standard. A Zwick Roell type Z020 device from Zwick GmbH & Co. KG, Ulm, Germany was used for the test. Three samples of each material were tested. The measurements were automatically recorded by the TestXpert program. Strength tests were performed both for the samples that were not tested in the bioreactor and for the samples tested in the bioreactor. In the case of paddles immersed in the activated sludge chamber and used for microbiological tests, they were washed before testing, dried for 24 h at 60 °C, cooled to room temperature, and then strength tests were performed.

### 2.6. Impact Bending Test

Impact bending tests of samples with a notch according to Charpy were performed in accordance with the PN-EN ISO 148-1:2017-02 standard [40], using INSTRON CEAST 9050. Due to technical reasons, this test was performed starting from the 61st day of immersion in the bioreactor to day 328. The tests were carried out in the shape of a cuboid with a size of 55 × 10 × 4 mm with a V-shaped notch with an opening angle of 45 ° and a depth of 2 mm. Six samples were tested.

### 2.7. Exposure of Samples in the Bioreactor

Initially, the studied samples were suspended in the bioreactor in the activated sludge aeration (nitrification) chamber, approx. 0.5 m below the surface of the sewage level in the mechanical and biological sewage treatment plant with increased removal of nutrients, which receives approx. 260,000 m^3^ of sewage per day from a large city in central Poland (Figure 2).

During approximately 14.5 months of exposure of the samples in the bioreactor, samples were collected five times for testing. During this process, five paddles from each batch of composite (A, B, C, D) were removed and separately placed in sterile bottles in order to allow transport to the laboratory for testing. In addition, during the days of sample collection, activated sludge in which the paddles were immersed was also collected from the bioreactor. The dates of sample collection for testing are shown in Table 2.

### 2.8. Microbiological Studies

The biofilm (precipitate formed on the samples—paddles) was thoroughly scraped from the samples collected for testing into sterile weighing bottles, and then each sample was rinsed twice with 2 mL of sterile water. The water together with the sediment suspension was poured into the vessels with the previously scraped biofilm. This method of collecting microorganisms guarantees that all forms of organisms will be detached from the sample for qualitative and quantitative tests. Microscopic preparations were obtained using the biofilm and were viewed under the Delta Optical Evolution 100 Trino Plan microscope from Delta Optical LLC, sp. k., Mińsk Mazowiecki, Poland. Observations were conducted at magnifications of 40×, 100× and 400× and/or 64×, 160× and 640×. The number of present microorganisms was determined on the basis of an estimation method based on a six-point scale (0—none, 1—single, 2—not very numerous, 3—quite numerous, 4—numerous, 5—very numerous), and their determination (systematic affiliation) was carried out [41,42,43,44]. Additionally, the presence of microorganisms in the activated sludge, which was collected on the same days as the collection of samples—paddles, was also analysed.

### 2.9. Determination of Biofilm Dry Matter

The residual sediment collected for microbiological analysis was used to determine the dry mass of the biofilm. The contents of the transport bottles were poured into weighed glass evaporators, and the residue from the transport bottles was thoroughly rinsed with sterile water and poured into the evaporators. The evaporators were then placed in a hot water bath to evaporate the water. After evaporation of water, the evaporators were placed in a dryer for 24 h at 60 °C. After this process, the evaporators were cooled in a desiccator and weighed using an analytical balance. The process of drying and weighing was repeated until a constant mass was obtained.

## 3. Results

The tested samples, which can be defined as biocomposites based on their composition, were analysed both before and after exposure in the bioreactor.

### 3.1. Density

The results of density measurements for the initial samples (before they were placed in the bioreactor) are summarized in Table 3. The lowest average density was observed in the case of sample A (pure polyethylene PE-HD), and the highest in sample D (60% PE-HD + 40% SCG). The increase in the density of composites is the result of using a filler with a higher density than that of polyethylene. The tests were carried out with the use of five test samples.

### 3.2. Water Content

The tested composites contained a low amount of water, which did not exceed 0.13% by weight, as is shown in Figure 3. The amount of water in the composites increases with the increase in SCG content. This most likely results from the introduction of water bound in the structure of the filler particles with the filler.

### 3.3. Water Absorption

The results of the water absorption test are shown in Figure 4.

Based on the obtained test results, it can be concluded that paddles that consist of various composites do not absorb a significant amount of water. The mass change of the analysed samples in any material did not exceed 1%. This means that the samples will not sink to the bottom of the bioreactor. The highest water absorption was observed for sample D (60% PE-HD + 40%SCG) and it amounted to 0.90%, while the lowest occurred for sample A (100% PE-HD) and it amounted to 0.16%. The tests showed that the water absorption increased with the percentage of coffee grounds contained in the composite. Hydrophilic filler particles have been introduced into the polyethylene matrix, which means that the water absorption increases with the increase in their content.

### 3.4. Contact Angle

The results of the contact angle measurements for the samples, depending on the composition and residence time in the bioreactor, are shown in Figure 5.

The test results indicate that the values of the contact angle for samples not introduced into the bioreactor, regardless of their composition, are similar and oscillate at approx. 60°, i.e., their surface is hydrophilic. In the case of all samples placed in the bioreactor, an increase in the contact angle was observed, with lower values of approx. 76.4° for sample A (PE-HD) and higher values of approx. 90° for the remaining samples. A significantly greater increase in the contact angle for composites after exposure in the bioreactor suggests that notable changes occurred on their surface during exposure. This indicates the formation of a biofilm.

### 3.5. Tensile Strength Tests of Composites

Based on the results of static tensile tests, the mechanical properties of the paddles were determined. The obtained results of the variability of Young’s modulus value of the control and test paddles in the bioreactor (collection dates 1–5) are presented in Figure 6.

The studies indicate that the value of Young’s modulus of control samples increases with the content of spent coffee grounds in the composite. The lowest modulus value was 212 MPa for pure polyethylene paddles (sample A), and the highest (332 MPa) for polyethylene paddles with 40 of SCG (sample D). Studies have shown that the value of Young’s modulus of the tested samples is usually lower compared to the samples not tested in the bioreactor. No changes in Young’s modulus values were observed for sample A (100% PE-HD) after tests in the bioreactor for 11 months. The highest value of Young’s modulus was obtained for sample D after 11 months in the bioreactor. This indicates that the presence of microorganisms during continuous immersion in the bioreactor liquid facilitates the increase in the stiffness of these composites.

### 3.6. Impact Bending Test

The results of the impact bending test from the second to the fifth sampling are presented in Figure 7. These tests were not conducted previously for technical reasons.

Based on the results of the impact strength of the tested samples, it can be concluded that sample A, i.e., 100% PE-HD, exhibits the highest resistance to dynamic impacts (average 51.82 kJ/m^2^). Its value was 3.6 to 6.6 times higher on average compared to other composites. Sample B exhibits the highest impact strength of the samples filled with spent coffee grounds (20% SCG), its average impact strength value was at 14.53 kJ/m^2^, and sample D (40% SCG) was the least resistant to mechanical impacts, with an average impact strength value of 7.89 kJ/m^2^. Based on the obtained results, it can be concluded that the impact strength of the composites decreases with the increase in spent coffee ground content and that the filler particles play the role of notches in these materials.

Studies regarding the use of waste from coffee production as components of biocomposites have been conducted for many years. Garcia-Garcia et al., 2015 [45], reported that adding coffee grounds to a polypropylene matrix causes a slight decrease in flexural strength and reduces deformations due to the phenomenon of stress concentration caused by dispersed particles in the PP matrix. At the same time, the flexural modulus increases as a result of a significant decrease in deformation capacity. Kufel and Kuciel 2019 [46] studied polypropylene (PP) composites with a 12.5% addition of ground coffee beans (powder filler) and observed a decrease in impact strength, a slight decrease in tensile strength, an increase in water absorption and stiffness, an increase in the modulus of elasticity, and the presence of a natural fragrance. Similar observations were reported by Tan et al., 2017 [47] regarding an increase in the tensile strength of the composite with the addition of ground coffee waste, higher impact properties, and improved water resistance. The use of coffee grounds as a filler for biodegradable polymeric materials can significantly reduce their amount in landfills and obtain a new material that will be biodegradable (Lebedew et al., 2021 [27]). In turn, the addition of high amounts of a natural filler (coffee husks) can negatively affect the tensile strength and elongation at break but increases the crystallinity of eco-composites, and thus their Young’s modulus and hardness. At the same time, it reduces the amount of produced greenhouse gases [48].

### 3.7. Activated Sludge Research

The basic physical parameters of the activated sludge from the nitrification chamber expressed as average values of measurements from the entire research period are summarized in Table 4.

The parameters of the activated sludge from all sample collection dates were very similar. This is associated with the operation of the sewage treatment plant, which receives high amounts of domestic and economic wastewater with equal parameters. In the analysed aeration chamber, there is a high content of total suspended solids, predominantly of organic origin. The concentration of the sludge in the reactor should not exceed 4000 mg/L, as the excess of suspended solids may pass to further stages of treatment together with the treated sewage. This is most often the case when the sedimentation properties of the sludge deteriorate. In the analysed case, the Mohlmann index of the activated sludge was high and averaged at 172.5 mL/g, while the sedimentation rate deteriorated.

Based on the microscopic analysis of activated sludge, the presence of numerous microorganisms classified into the following taxa was established: Rhizopoda—*Rhizopoda*, Flagellates—*Flagellata*, Ciliata—*Ciliata*, Rotatorias—*Rotatoria*, Gastrotrich—*Gastrotricha*, Nematodes—*Nematoda n.det.*, Filamentous bacteria and fungi n.det.—*Filamentous bacteria et fungos* n.det., Free-floating bacteria n.det.—*Liberum natantes Bacteria* n.det. The most quantitatively and qualitatively diverse group was the *Ciliata*, among which there were sedentary, creeping, and free-floating forms. The estimated number of microorganisms present in the activated sludge in the aeration chamber is summarized in Table 5, while the average percentage share of the main microbial taxa present in the activated sludge during all six sample collections is shown in Figure 8.

Activated sludge from the fourth and third collection dates is characterized by the highest diversity of microorganisms (sum of points and genus or species), while sludge from the first and third collection dates achieved the highest results in terms of the presence of the main taxa. This was due to the additional occurrence of such taxa as *Gastrotricha* and *Nematoda*. Exposure to more SCG is likely to promote microbial diversity.

In turn, during the last two samplings, the activated sludge was of the worst quality, the sludge flocs were small and single, the diversity of taxa as well as the microbial abundance were relatively low, and their mobility was also limited. This could be due to sludge overload or problems with reactor oxygenation.

### 3.8. Microbiological Tests of Composites

Paddles with different compositions (samples A, B, C, D) immersed in the activated sludge were used by the activated sludge microorganisms present in the bioreactor to form the biological membrane (biofilm). In order to investigate the susceptibility of individual paddles to the deposition of biological membranes and microorganisms present on their surface, microscopic examinations of the biofilm from the collected paddles were performed. The estimated numbers of microorganisms present on individual paddles in all research periods are summarized in Table 6.

In the scope of all sample collection dates, the highest diversity in terms of the presence of microorganisms (sum of points) was found in paddles consisting of 60% PE-HD + 40% spent coffee grounds (samples D), followed by samples C (70% PE-HD + 30% spent coffee grounds). At the same time, the lowest diversity of microorganisms was observed for a series of sample A, which consisted of pure polyethylene (100% PE-HD).

With the increase in the SCG content in the composites, the degree of their surface development increases, as shown in Figure 1. The greater the degree of surface development, the greater the number of microorganisms on the surfaces of the samples.

Based on the obtained test results, it can also be concluded that the number of detected taxa of microorganisms that form the biological film on the surface of all samples was susceptible to fluctuations during their immersion in the bioreactor. The numbers of taxa occurring on individual composite samples for all sample collection dates are compared in Figure 9.

After an analysis of the number of taxa detected on individual samples, it can be concluded that samples D dominated in terms of diversity, with the exception of the fourth and fifth sampling. In the fourth sampling, more taxa were detected in the case of sample A (100% PE-HD), while sample C (70% PE-HD + 30% SCG) achieved the best results in the fifth sampling.

At the same time, research data show that in most cases the increase in the ratio of spent coffee grounds in the composite results in increased diversity and abundance of microorganisms. The most numerous microorganisms found in all samples were sessile ciliates belonging to the genus *Epistylis*. Their number on the paddles was higher compared to their presence directly in the activated sludge chamber. This may confirm the beneficial effect of spent coffee grounds on the deposition of this type of ciliates. The sum of points obtained from the estimated number of microorganisms detected on individual composites and the total point value calculated from all samples is presented in Table 7.

The obtained point values indicate that composite D (with the composition of 60% PE-HD + 40% spent coffee grounds) is the best and most susceptible filler for the deposition of microorganisms, and pure polyethylene (100% PE-HD) is the least susceptible. Sample C (70% PE-HD + 30% coffee grounds) also exhibited good conditions for the settling of microorganisms.

### 3.9. Biofilm Dry Matter

The average dry matter of the biofilm calculated per a single paddle from all five samplings reached the highest value of 0.1560 g in the case of composite C (70% PE-HD + 30% SCG), while the lowest value of 0.0778 g was observed in case of sample A (100% PE-HD HD). In the case of the remaining samples, the biofilm dry matter values were similar; for sample B (80% PE-HD + 20% coffee grounds) the average value was equal to 0.1010 g, while in the case of sample D (60% PE-HD + 40% SCG) it was 0.1036 g. The average dry matter of the biofilm calculated per a single paddle at particular dates of sampling is presented in Figure 10.

The amount of dry matter of the biofilm fluctuated significantly with respect to particular dates of sampling. The highest values of dry matter obtained in the fifth series of tests correspond with the period when the lowest number of taxa was recorded in the activated sludge (20 genera or species). The calculated number of microorganisms was equal to 29 and it was also the lowest among all sampling periods (Table 5). The lowest values of dry matter were found during the second sampling, during which the presence of 24 taxa was found in the activated sludge, and the point value of the number of microorganisms was equal to 44. Therefore, it can be assumed that the process of biofilm formation and its possible leaching (removal from the surface of paddles) or deposition of activated sludge flocs is a dynamic process that changes over time and may be related to the intensity of wastewater flow, the process of aeration, the composition of microorganisms present in the activated sludge as well as the temperature. For this reason, the final fittings suspended in the moving bed-chamber must be characterized by a spatial, multidimensional structure, with a large specific surface area and be protected against the processes of abrasion of the biological membrane during sludge mixing and mutual contact of the outer surface of the fittings.

## 4. Conclusions

Currently, composites play an important role in all aspects of engineering and technology with constantly growing applications. Recently, more attention was focused on natural fillers due to their suitability as reinforcement materials in thermoplastic matrices which improve the mechanical properties of these polymers. Biofillers are used due to their low cost, high strength rigidity, non-toxicity, biodegradability, and availability. Currently, spent coffee grounds (SCG) are attracting more attention as a natural filler since high amounts of SCG are generated every day (food waste of coffee processing). This study allowed us to determine the long-term effect of activated sludge microorganisms with known technical and technological parameters on the mechanical properties of composites with spent coffee grounds filler. The research showed a direct effect of the filler on the modulus of elasticity of the composites and on the susceptibility of their surface to biofilm formation. Materials containing 30 and 40% of the filler were composites with high resistance to deformation. Their Young’s modulus was higher by 50 and 57%, respectively, compared to the PE-HD modulus. It was found that the initial hydrophilicity of the material, which determines its high wettability, is a contributing factor to the formation of biofilm and results in good adhesion of microorganisms to the surface of the used fittings, thus enabling their colonization. The biofilm formed on the surface of the tested materials increased the contact angle. An increase in the contact angle was observed in all samples tested in the bioreactor, with lower values of approx. 76.4° for sample A (PE-HD) and higher values of approx. 90° for the remaining samples of composites with spent coffee grounds filler. The research confirmed that the increased ratio of coffee grounds in the composite results in an increase in the diversity and number of microorganisms. The highest number and the greatest diversity of microorganisms were recorded in the case of the composite with 40% of coffee grounds during more than a year of exposure in the bioreactor, while the composite with 30% SCG ranked second. Ciliates, especially sedentary forms belonging to the *Epistylis* genus, were the most common and the most numerous groups of microorganisms in the activated sludge and in the biofilm observed on samples after immersion in the bioreactor. Protists play an important role in activated sludge, primarily as consumers of numerous bacteria and small organic particles, thus contributing to the rapid biodegradation of organic pollutants. The highest biofilm dry matter value was found on the composite which consisted of 70% PE-HD + 30% coffee grounds, and the lowest on the pure polyethylene mouldings. The conducted research confirms that the use of polymer composite mouldings with a filler in the form of spent coffee grounds as a carrier allows for the efficient increase in the population of microorganisms in the bioreactor. Due to the well-developed active surface of the composite carrier, the sensitivity of the population of microorganisms to unfavourable environmental conditions is reduced.

## Figures and Tables

**Figure 1 materials-16-01181-f001:**
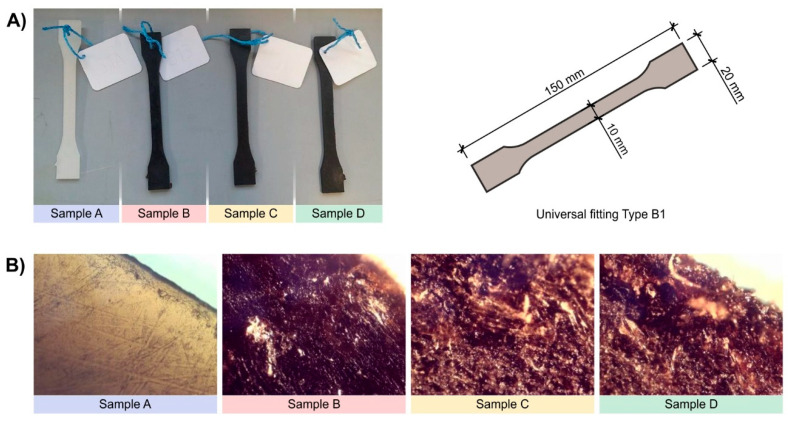
(**A**) Samples (paddles) of high-density polyethylene (PE-HD) and with the addition of SCG with a nameplate, (**B**) Microscopic images of test paddles, magnification 160 times.

**Figure 2 materials-16-01181-f002:**
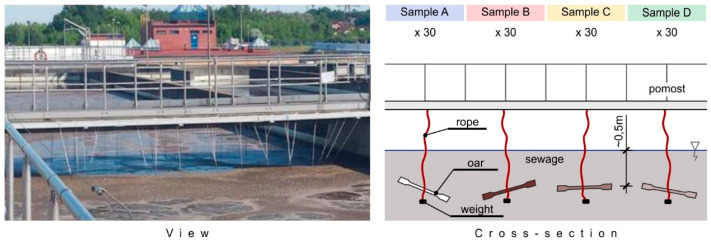
Samples suspended in the bioreactor (in the nitrification chamber).

**Figure 3 materials-16-01181-f003:**
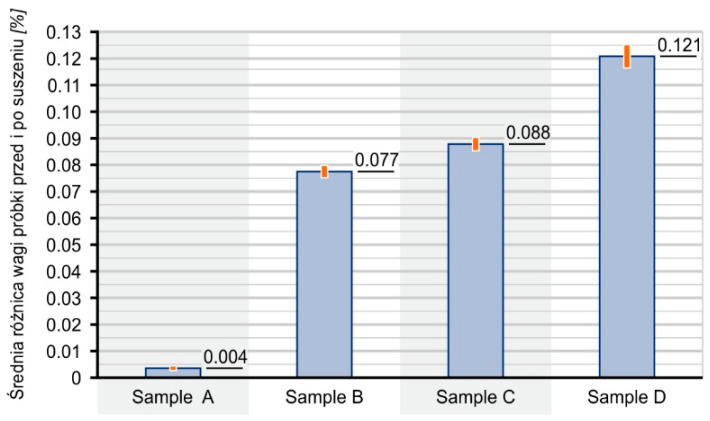
Water content in samples before testing in the bioreactor.

**Figure 4 materials-16-01181-f004:**
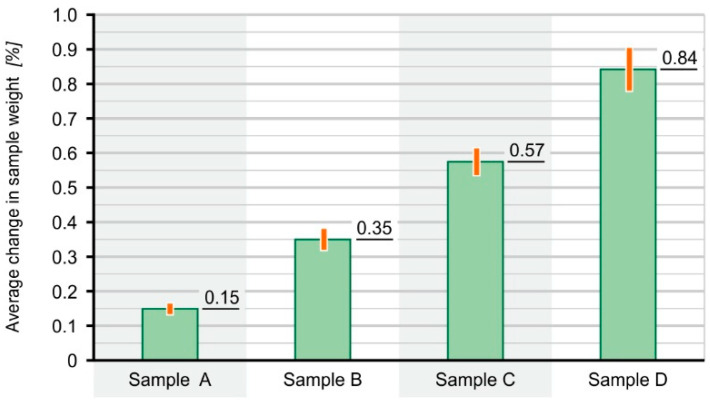
Results of water absorption by composite samples.

**Figure 5 materials-16-01181-f005:**
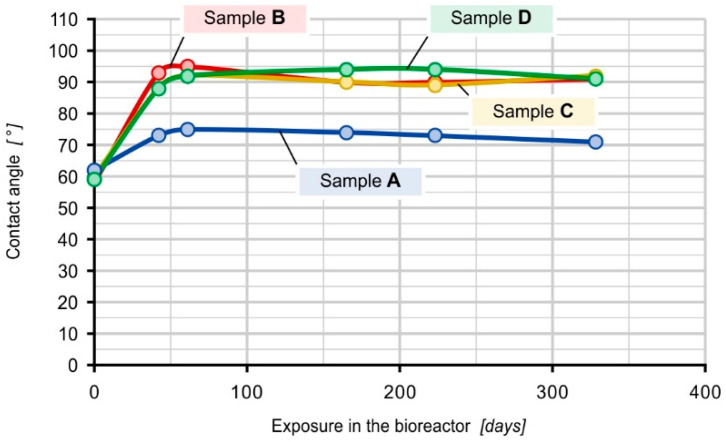
Contact angle values for reference samples and samples placed in the bioreactor.

**Figure 6 materials-16-01181-f006:**
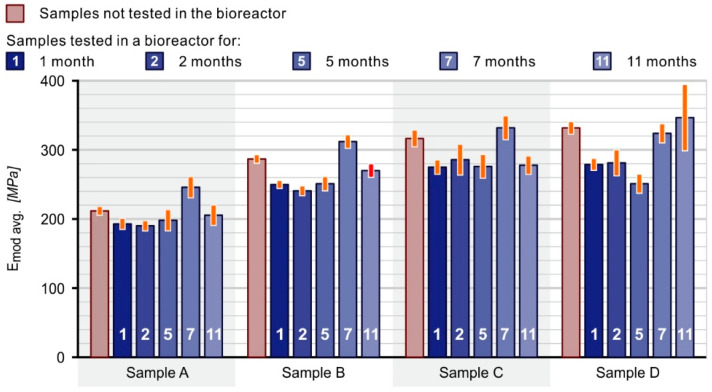
Young’s modulus values of reference and reactor tested paddles.

**Figure 7 materials-16-01181-f007:**
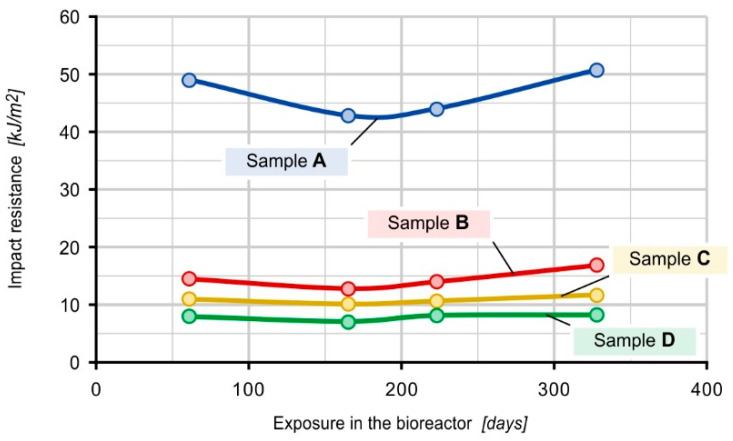
Impact bending test results.

**Figure 8 materials-16-01181-f008:**
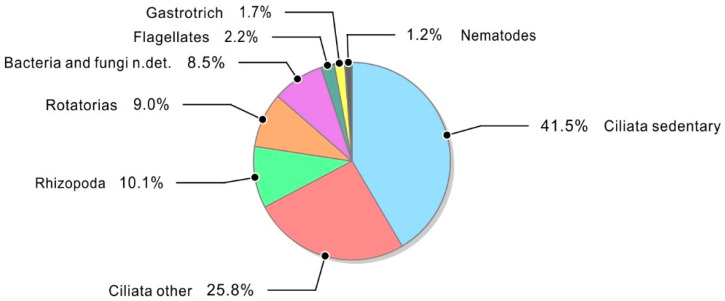
Average percentage of taxa in the activated sludge from the aeration chamber.

**Figure 9 materials-16-01181-f009:**
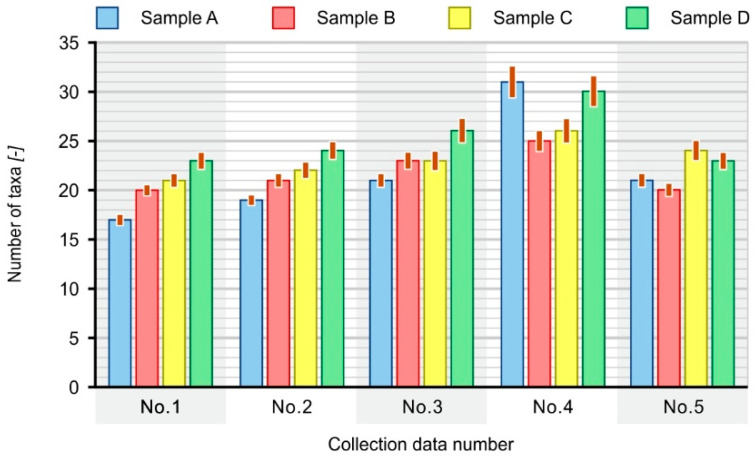
Number of microbial taxa detected on the paddles (A, B, C, D) during the study.

**Figure 10 materials-16-01181-f010:**
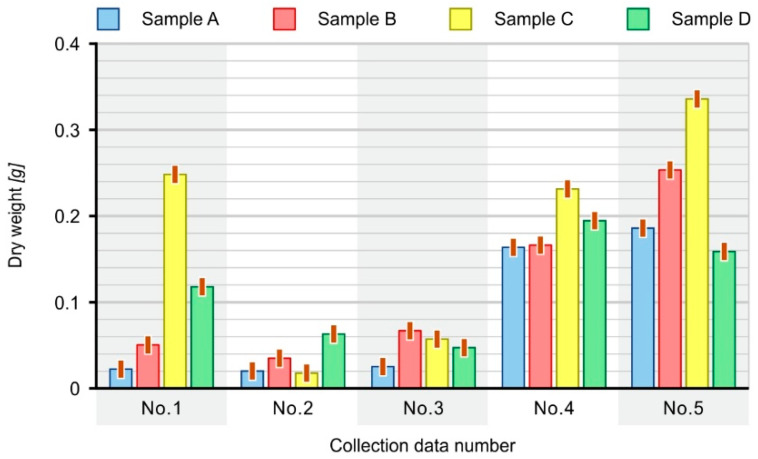
Average dry matter of biofilm on composites in successive dates (numbers) of sampling.

**Table 1 materials-16-01181-t001:** Composition of the tested samples.

Type of Sample	Matrix PE-HD [mass %]	Type of Filling	Amount of Filler [mass %]
A	100	-	0
B	80	Spent coffee (SCG)	20
C	70	Spent coffee (SCG)	30
D	60	Spent coffee (SCG)	40

**Table 2 materials-16-01181-t002:** Exposure times of test samples.

Type of Sample	Exposure in the Bioreactor
0	Placing samples in a bioreactor
1	After 42 days
2	After 61 days
3	After 165 days
4	After 223 days
5	After 328 days

**Table 3 materials-16-01181-t003:** Density of the tested samples.

Type of Sample	Sample Density [g/cm^3^]
A	0.9021 ± 0.0442
B	0.9527 ± 0.0380
C	0.9856 ±0.0334
D	1.0162 ± 0.4212

**Table 4 materials-16-01181-t004:** Parameters of activated sludge.

Parameter	Unit	Average Value
pH	-	7.35
Total suspension	mg/L	5540
Mineral suspension	mg/L	1184
Organic suspension	mg/L	4356
Settling after 30 min.	mL/1000 mL	960 + gas
Settling after 60 min.	mL/1000 mL	890 + gas
Settling after 120 min..	mL/1000 mL	750 + gas
Activated sludge index	mL/g	172.5

**Table 5 materials-16-01181-t005:** Estimated number of microorganisms in activated sludge.

Microorganisms	Genus or Species	Estimated Number of Microorganisms				
		Sampling Number				
		No. 1	No. 2	No. 3	No. 4	No. 5
*Rhizopoda*	*Arcella vulgaris*	3	3	1	4	2
	*Amoeba proteus*	1	1	1	1	0
*Flagellata*	*Peranema trichophorum*	0	1	1	1	1
	*Bodo* sp.	0	0	0	1	0
*Ciliata*	*Carchesium polypinum*	3	1	3	2	0
	*Epistylis lacustris*	1	4	0	3	2
	*Epistylis plicatilis*	0	4	5	3	0
	*Epistylis rotans*	1	4	0	3	1
	*Epistylis coronata*	0	4	3	3	0
	*Epistylis chrysemydis*	0	4	3	2	0
	*Opercularia coarctata*	1	1	0	3	2
	*Vorticella campanula*	0	1	3	2	0
	*Vorticella convallaria*	0	1	3	2	2
	*Vorticella microstoma*	3	1	3	2	1
	*Acineta tuberosa*	1	0	1	1	1
	*Podophrya fixa*	1	0	0	0	0
	*Tokophrya infusionum*	1	1	1	1	1
	*Tokophrya lemnarum*	0	1	1	0	1
	*Aspidisca* sp.	5	2	3	2	2
	*Spirostomum* sp.	1	1	1	1	0
	*Euplotes* sp.	1	0	0	2	1
	*Litonotus* sp.	1	1	2	1	1
	*Glaucoma scintillans*	1	1	1	1	2
	*Paramecium bursaria*	1	1	1	3	1
	*Stentor* sp.	0	0	1	0	0
	*Chilodonella* sp.	0	0	0	0	1
	*Colpidium colpoda*	0	0	0	1	0
	*Oxytricha* sp.	0	0	0	0	1
	*Discophyra elongata*	0	0	0	0	0
	*Paramecium caudatum*	0	0	1	2	0
*Rotatoria*	*Rotaria rotatoria*	2	2	1	3	1
	*Lecane* sp.	1	1	1	1	0
	*Cephalodella* sp.	0	0	1	1	0
*Gastrotricha*	*Gastrotricha n.det.*	1	0	1	0	0
*Nematoda*	*Nematoda n.det.*	1	0	0	0	0
*Bacteria and Fungi*	Filamentous bacteria and fungi *n.det*	2	1	2	2	3
*Bacteria*	Free swimming bacteria *n.det*	2	2	2	3	2
**Sum of points** **Number of taxa**		**35**	**44**	**47**	**57**	**29**
		**22**	**24**	**26**	**29**	**20**

Signs: 0—none; 1—single; 2—not very numerous; 3—quite numerous; 4—numerous; 5—very numerous.

**Table 6 materials-16-01181-t006:** Estimated number of microorganisms on the paddles.

Microorganisms	Genus or Species	Estimated Number of Microorganisms																			
		Sampling Number																			
		No. 1				No. 2				No. 3				No. 4				No. 5			
		Sample				Sample				Sample				Sample				Sample			
		A	B	C	D	A	B	C	D	A	B	C	D	A	B	C	D	A	B	C	D
*Rhizopoda*	*Arcella vulgaris*	1	1	4	1	1	1	2	2	0	1	1	1	1	0	2	2	1	1	1	1
	*Amoeba proteus*	0	0	0	1	1	1	0	1	1	0	0	0	0	0	0	0	0	0	0	0
*Flagellata*	*Peranema trichophorum*	0	1	1	1	0	1	1	1	1	1	1	1	1	2	1	1	0	1	2	0
	*Bodo* sp.	1	1	1	1	1	0	0	0	0	1	0	1	1	0	1	1	0	0	1	1
*Ciliata*	*Carchesium polypinum*	1	1	1	1	1	1	0	1	0	0	0	0	1	1	0	1	1	0	1	1
	*Epistylis lacustris*	5	5	5	5	4	5	5	5	4	5	5	5	4	5	5	5	5	5	5	5
	*Epistylis plicatilis*	5	5	5	5	4	5	5	5	4	5	5	5	4	5	5	5	4	5	5	5
	*Epistylis rotans*	5	5	5	5	4	5	5	5	4	5	5	5	4	5	5	5	5	5	5	5
	*Epistylis coronata*	5	5	5	5	4	5	5	5	4	5	5	5	4	5	5	5	4	5	5	5
	*Epistylis chrysemydis*	5	5	5	5	4	5	5	5	5	5	5	5	4	5	5	5	4	5	5	5
	*Opercularia coarctata*	1	2	0	0	0	1	1	1	1	0	5	4	2	3	3	3	1	2	2	2
	*Vorticella campanula*	0	0	0	1	0	0	1	1	0	0	0	0	1	0	1	0	0	0	1	0
	*Vorticella convallaria*	1	0	1	0	1	1	0	0	0	0	0	1	1	1	1	1	1	0	1	2
	*Vorticella microstoma*	1	1	1	1	1	1	1	1	1	0	0	0	1	1	0	1	0	2	0	0
	*Trichodina pediculus*	1	1	1	1	0	0	1	0	0	1	0	0	0	0	1	0	1	0	1	3
	*Acineta tuberosa*	0	0	0	1	0	0	0	0	1	1	1	1	1	0	0	1	0	0	0	0
	*Acineta flava*	0	1	0	0	0	0	1	0	0	1	1	1	0	0	0	0	0	0	0	0
	*Tokophrya infusionum*	1	1	1	1	1	1	1	1	2	1	1	2	1	1	1	1	1	1	0	0
	*Tokophrya lemnarum*	0	0	0	0	0	1	1	1	2	1	1	2	1	1	0	1	1	1	0	0
	*Aspidisca costata*	1	2	2	3	1	1	2	3	1	1	1	1	1	1	0	1	1	2	1	1
	*Chilodonella cucullulus*	0	0	0	0	0	0	0	0	0	1	0	0	0	0	0	1	0	0	0	0
	*Litonotus* sp.	0	1	1	1	1	1	1	1	2	1	1	1	1	1	1	1	0	1	1	1
	*Glaucoma scintillans*	0	1	0	1	0	1	1	2	1	1	0	1	1	1	1	1	2	2	2	2
	*Paramecium caudatum*	0	0	0	0	0	1	0	1	1	0	1	1	1	2	2	1	1	1	1	1
	*Paramecium bursaria*	0	0	1	1	1	0	1	1	1	1	1	1	0	0	0	0	0	0	0	0
	*Stylonychia mytilus*	0	0	0	0	0	0	0	0	1	0	0	0	1	0	0	3	0	0	1	2
	*Discophyra elongata*	0	0	0	0	0	0	0	0	0	0	1	3	1	1	1	0	0	0	0	0
	*Thuricola folliculata*	0	0	0	0	0	0	0	0	0	0	0	0	1	1	1	0	0	0	1	0
	*Euplotes affinis*	0	0	0	0	0	0	0	0	0	0	0	1	0	0	0	0	0	0	0	0
	*Spirostomum* sp.	0	0	0	0	0	0	0	0	0	0	0	1	1	1	1	0	0	0	0	0
	*Colpidium colpoda*	0	0	0	0	0	0	0	0	0	0	0	0	1	1	0	1	0	0	0	0
	*Amphileptus daparedei*	0	0	0	0	0	1	1	0	0	0	0	0	0	0	0	1	0	0	0	0
	*Stentor* sp.	0	0	0	0	0	0	0	0	0	0	0	0	0	0	0	0	1	0	0	1
	*Oxytricha* sp.	0	0	0	0	0	0	0	0	0	0	0	0	1	1	1	1	1	1	1	1
*Rotatoria*	*Rotaria rotatoria*	1	1	1	1	1	0	1	1	1	1	1	1	1	1	2	1	4	4	4	4
	*Lecane* sp.	0	1	0	0	0	0	0	1	0	1	0	0	1	0	1	1	0	0	0	0
	*Cephalodella* sp.	0	0	1	1	0	0	0	0	0	1	1	0	0	0	0	0	0	0	0	0
*Oligochaeta*	*Tubifex tubifex*	0	0	0	0	1	0	0	0	0	0	0	0	-	-	-	-	-	-	-	-
	*Aeolosoma* sp.	0	0	0	0	0	0	0	0	0	0	1	1	0	0	1	1	0	0	0	0
*Gastrotricha*	*Gastrotricha n. det.*	0	0	0	0	0	0	0	0	0	0	0	0	1	0	1	1	1	1	1	1
*Nematoda*	*Nematoda n. det.*	0	0	1	0	0	0	0	1	0	0	1	0	0	0	0	0	0	0	0	1
*Bacteria and Fungi*	Filamentous bacteria and fungi *n.det.*	1	1	1	1	1	1	1	2	1	2	2	2	2	2	2	2	2	2	1	2
*Bacteria*	Free—swimming bacteria *n.det.*	2	2	2	2	2	2	2	3	1	2	3	2	3	4	4	4	2	3	3	3
**Sum of points**		**38**	**44**	**46**	**46**	**35**	**42**	**45**	**51**	**40**	**45**	**50**	**55**	**50**	**52**	**55**	**59**	**44**	**50**	**53**	**55**
**Number of taxa**		**17**	**20**	**21**	**23**	**19**	**21**	**22**	**24**	**21**	**23**	**23**	**26**	**31**	**25**	**26**	**30**	**21**	**20**	**24**	**23**

Signs: 0—none; 1—single; 2—not very numerous; 3—quite numerous; 4—numerous; 5—very numerous.

**Table 7 materials-16-01181-t007:** Cumulative sum of points obtained from samples during all collections.

Type of Sample	Points					
	Sampling Number					
	No. 1	No. 2	No. 3	No. 4	No. 5	Sum of Points
A	38	35	40	50	44	246
B	44	42	45	52	50	274
C	46	45	50	55	53	292
D	46	51	51	59	55	312

## Data Availability

Not applicable.

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
