# Peer review of "Spent Coffee as a Composite Filler for Wastewater Treatment"

_materials, 2023, doi:10.3390/ma16031181_

Round 1

Reviewer 1 Report

This paper is ready for publication in journals with the following moderate modifications.

1.        The novelty and advantages of this spent coffee as a filling of composites should be highlighted and compared with previous research.

2.        “Water absorption” should be “Water adsorption”?

3.        Table 3, how to get the error bar value? Please indicate the times of parallel test.

4.        The advances of wastewater treatment technology should be enriched to enhance the depth of the article, e.g.: Chinese Journal of Catalysis, 2022, 43, 2652–2664; Adv. Fiber Mater., 2022, https://doi.org/10.1007/s42765-022-00189-w; J. Colloid Interface Sci., 624 (2022) 219-232;

5.        How about the durability of this spent coffee as a filling of composites?

6.        The possible reason for the increased ratio of coffee grounds in the composite results in an increase of the diversity and number of microorganisms should be analyzed and discussed in detail.

7.        There are many grammatical errors and unclear statement in the text, please carefully check and modify.

Author Response

Dear Reviewer,
We would like to thank the reviewer for careful and throughout reading of this manuscript and for the thoughtful comments and constructive suggestions, which help us to improve the quality of this manuscript. We also thank the reviewer for the effort and time that was put into the review of the manuscript 
We have carefully reviewed the comments and have revised the manuscript accordingly. Our responses are given in a point by point manner below.
1. The novelty and advantages of this spent coffee as a filling of composites should be highlighted and compared with previous research.

Answer: The introduction was changed and supplemented with additional information and  references.

2.  “Water absorption” should be “Water adsorption”?

Answer: The correct version is “Water absorption”. It is the amount of water absorbed in the composites was calculated by the weight difference between the samples exposed to water and the zero sample before exposed to water.

3.  Table 3, how to get the error bar value? Please indicate the times of parallel test.

Answer: We thank the reviewer for the suggestion, we changed and we completed informtion.

4. The advances of wastewater treatment technology should be enriched to enhance the depth of the article, e.g.: Chinese Journal of Catalysis, 2022, 43, 2652–2664; Adv. Fiber Mater., 2022, https://doi.org/10.1007/s42765-022-00189-w; J. Colloid Interface Sci., 624 (2022) 219-232;

Answer: The citation of the literature from which this citation was taken has been added Chinese Journal of Catalysis, 2022, 43, 2652–2664; Adv. Fiber Mater., 2022, https://doi.org/10.1007/s42765-022-00189-w; J. Colloid Interface Sci., 624 (2022) 219-232

5.  How about the durability of this spent coffee as a filling of composites?

Answer: SCGs are thermally stable up to about 200oC. Spent coffee is a mixture of polysaccharides and aromatic polymers. Their chemical structure is similar to that of natural fillers such as wood. This allows us to believe that when used as elements of a moving bed, they will fulfill their function for a long time, similarly to wood-based fillers.

6.  The possible reason for the increased ratio of coffee grounds in the composite results in an increase of the diversity and number of microorganisms should be analyzed and discussed in detail. 

Answer: The citation of the literature has been added M. AuguÅ›cik-Królikowska, J. Ryszkowska, A. Ambroziak, L. Szczepkowski, R. Oliwa, M. Oleksy; The structure and properties of viscoelastic polyurethane foams with fillers from coffee grounds; Polimery 2020, 65, 10, 708-718 , The article has supplemented the discussion.

7.        There are many grammatical errors and unclear statement in the text, please carefully check and modify. 

Answer: We thank the reviewer for the suggestion, we chacked and modified it.

We are grateful for your consideration of this manuscript, and we also very much appreciate your suggestions, which have been very helpful in improving the manuscript.  All the comments we received on this study have been taken into account in improving the quality of the article.
We believe that the reviewers’ suggestions have been very helpful in improving this manuscript. We hope that these changes to the manuscript will facilitate the decision to publish this study . In any case, we are open to consideration of any further comment on our answers.

Reviewer 2 Report

This study tested the susceptibility of the surface of composite fittings to the formation of a biofilm in the wastewater treatment process. The fittings consisted of high-density polyethylene (PE-HD) used as the matrix and a filler from spent coffee grounds (SCG) was used as a modifier. I recommend the manuscript for publication after considering the following suggestions which their addressing will fit the manuscript for publication.

Comments

1.     More profound discussions and comparisons with other published works are welcomed. It is pertinent to compare the findings with other methods and show how the study is novel compared to others.

2.     Procedure followed in the experimental section must be supported by references.

3.     The Manuscript needs thorough revision to improve the text quality and readability of work.

4.     Please check the grammar, uniformity in the reference format, and spell-check are necessary throughout the manuscript. 

Author Response

Dear Reviewer,
We would like to thank the reviewer for careful and throughout reading of this manuscript and for the thoughtful comments and constructive suggestions, which help us to improve the quality of this manuscript. We also thank the reviewer for the effort and time that was put into the review of the manuscript 
We have carefully reviewed the comments and have revised the manuscript accordingly. Our responses are given in a point by point manner below.

1.     More profound discussions and comparisons with other published works are welcomed. It is pertinent to compare the findings with other methods and show how the study is novel compared to others.

Answer: The abstract, introduction and conclusion was changed and supplemented with additional information and  references.

2.     Procedure followed in the experimental section must be supported by references.

Answer: We thank the reviewer for the suggestion, we changed and we corrected text and we supplemented with additional information and references. 

3.     The Manuscript needs thorough revision to improve the text quality and readability of work.

Answer: We are grateful for your consideration of this manuscript - your suggestions, which have been very helpful in improving the manuscript.  All the comments we received on this study have been taken into account in improving the quality of the article.

4.     Please check the grammar, uniformity in the reference format, and spell-check are necessary throughout the manuscript. 

Answer: We thank the reviewer for the suggestion, we changed and we corrected text.

We believe that the reviewers’ suggestions have been very helpful in improving this manuscript. We hope that these changes to the manuscript will facilitate the decision to publish this study . In any case, we are open to consideration of any further comment on our answers.

Reviewer 3 Report

Manuscript No: materials-2131820

Title: Spent coffee as a filling of composites for application in wastewater treatment technology.

Comments

Minor revision required

1.      More results should be explored in abstract and in conclusion section

2.      Novelty should be clearly defined in abstract, introduction and in conclusion section.

3.      In introduction line 99-104 discuss some latest adsorbent for the removal of pollutants from waste water like Inorganic Chemistry Communications 145 (2022) 110008, Surfaces and Interfaces 34 (2022) 102324.

4.      Authors claimed, SCG particles are characterized by a size in the range 13–890 μm. The median particle size is 221 μm.! How they calculate particle size. Justify

5.      Error bars should be provided for figure 9 and 10

6.      There are so many typo, grammatical errors in whole manuscript, should be revised by some native speaker and formatting should be checked.

Author Response

Dear Reviewer,

We would like to thank the reviewer for careful and throughout reading of this manuscript and for the thoughtful comments and constructive suggestions, which help us to improve the quality of this manuscript. We also thank the reviewer for the effort and time that was put into the review of the manuscript

We have carefully reviewed the comments and have revised the manuscript accordingly. Our responses are given in a point by point manner below.

  1. More results should be explored in abstract and in conclusion section.

Answer: The abstract was changed and supplemented with additional information.

  1. Novelty should be clearly defined in abstract, introduction and in conclusion section.

Answer: The abstract, introduction and conclusion was changed and supplemented with additional information and  references..

  1. Authors claimed, SCG particles are characterized by a size in the range 13–890 μm. The median particle size is 221 μm.! How they calculate particle size. Justify.

Answer: We thank the reviewer for the suggestion, we changed and we completed informtion. („Particle size of SCG was determined in a Horiba LA-950 laser particle size analyzer using the LALLS (Low Angle Laser Light Scattering) technique to achieve full geometric and morphological characteristics of powders. The measurements were carried out with the refractive index specific to cellulose (1.47) laser setting, which reports the full geometrical and morphological characteristics of powders. Detailed results of SCG research are presented in the paper : M. AuguÅ›cik-Królikowska, J. Ryszkowska, A. Ambroziak, L. Szczepkowski, R. Oliwa, M. Oleksy; The structure and properties of viscoelastic polyurethane foams with fillers from coffee grounds; Polimery 2020, 65, 10, 708-718”).

  1. Error bars should be provided for figure 9 and 10

Answer: Error bars in figure 9 and 10 were supplemented.

  1. There are so many typo, grammatical errors in whole manuscript, should be revised by some native speaker and formatting should be checked.

Answer: We thank the reviewer for the suggestion, we changed and we corrected text..

We are grateful for your consideration of this manuscript, and we also very much appreciate your suggestions, which have been very helpful in improving the manuscript.  All the comments we received on this study have been taken into account in improving the quality of the article.

We believe that the reviewers’ suggestions have been very helpful in improving this manuscript. We hope that these changes to the manuscript will facilitate the decision to publish this study . In any case, we are open to consideration of any further comment on our answers.

Round 2

Reviewer 1 Report

After checking the revised version, I think that the authors have well addressed the issues raised by the reviewer. In this case, this manuscript can be recommended for publication